# Assessing the predictive value of morphological traits on primary lifestyle of birds through the extreme gradient boosting algorithm

**Luis Javier Madrigal-Roca**[ID]¤*

Applied Genetic Group, Department of Plant Biology, Faculty of Biology, University of Havana, Vedado, Havana, Cuba

¤ Current address: Ecology and Evolutionary Biology's Department of the University of Kansas, Lawrence, Kansas, United States of America

* madrigalrocalj@yahoo.com

## Abstract

The relationship between morphological traits and ecological performance in birds is an important area of research, as it can help us to understand how birds are able to adapt and how they are affected by changes in their environment. Many studies have investigated the relationship between morphological traits and certain aspects of the performance and ecological niche of these animals. However, the relationship between morphological traits and the primary lifestyle of birds has not previously been explored. This paper aims to evaluate the predictive potential of morphological data to determine the primary lifestyle of birds through a tree-based machine learning algorithm. By doing this, it is also possible to evaluate these artificial categories that we used to split up birds and know whether they are suitable for dividing them in function of shared morphological characteristics or need a redefinition under more discriminant criteria. Supplementary dataset 1 of the AVONET project was used, which comprises the 11 morphological predictors used in this work and the classification according to the primary lifestyle for more than 95% of the existing bird species. For all morphological traits used, statistically significant univariate differences were found between primary lifestyles. The three fitted machine learning models showed high accuracy, in all cases above 78% and superior to the ones achieved through traditional approaches used as contrasts. The results obtained provide evidence that primary lifestyle can be predicted in birds based on morphological traits, as well as more insights about the relevance of functional traits for ecological modeling. This is another step forward in our mechanistic understanding of bird ecology, while exploring how birds have adapted to their environments and how they interact with their surroundings.

**Data Availability Statement:** Relevant data, including the R script, is available at https://doi.org/10.6084/m9.figshare.16586228.v7 and https://

**Funding:** The author received no specific funding for this work.

**Competing interests:** The author declares that no competing interests exist.

## Introduction

Trait and functional trait conceptualization have become a keystone in studies of community assembly, biodiversity conservation, dynamic, and species co-occurrence [1, 2]. A trait is a feature that can be measured in individual organisms or superior levels of organization. Functional traits is a particular distinction that refers to traits where they are linked in a direct way in studies to a function, although is a fuzzy term, because every trait almost always has functional implications at certain levels [2]. One decisive factor to the increasing interest in their use is that they can promote a more mechanistic comprehension of ecosystems' structure and function [2–4].

In many ecological approaches, the concept of Hutchinson's niche is a useful framework that allows embedding the species and their traits within an space of multiple dimensions [5–8]. Under this idea, scientists can potentially describe ecological units by independent axes that include, for instance, traits, resource needs, or tolerances [6]. The concept of Hutchinson niche can be linked to the prediction of lifestyles in birds according to their morphological traits by examining how the morphological traits of birds are related to their ecological roles and how these roles are related to their niches. In this context, the primary lifestyles defined in [4], could be potentially predicted using morphological data. This could be supported by the idea that these ecological guilds should group a set of species with significant overlap in multidimensional resource niche space [8].

The morphology of birds is essential for them to perform diverse ecological functions: foraging, locomotion, reproduction, and predation. Beak size and form, for instance, are narrowly associated with dietary and foraging strategies [9–11]. For example, granivorous species generally have ridges on the edge of beaks that facilitate seed cracking, and also a suitable maxillary beak–skull articulation that contributes to cushioning the shock [12]. On the other hand, wing features align with flight style and energy preservation [13]. For example, some studies show that for certain birds narrow wings perform best for gliding and soaring [14]. Other traits, such as body size, are relevant too when analyzing the ecology of birds. Bigger birds tend to have lower metabolic rates than smaller ones [15, 16], an element that can limit the dispersion ability and promote terrestrial foraging strategies [17]. Moreover, in passerine birds, morphological traits such as bill size and shape, tarsus length, and wing pointedness were significantly associated with ecological characters such as diet, habitat, and migration [18].

Some ecomorphological studies have shown that, perhaps, trait combinations could be applied to construct niche classification models. This can be accomplished using classical methods like multinomial regressions and linear discriminant analysis. However, algorithms of machine learning capable of extracting insights from massive amounts of data have become important tools in this context. Two fundamental reasons justify the power of this approach, the usage of effective mathematical models to capture complex data relations and the existence of scalable learning systems [19]. However, there always have been limitations regarding restricted taxonomic scope, spatial scales, and coverage and completeness, which have limited consensus regarding structure or generalizations about morphofunctional links in animals [3, 7].

Recent analysis with big datasets of birds has found that morphological features can powerfully predict trophic niches using machine learning algorithms [7]. Moreover, [4] found evidence that morphological traits are also indicatives of geographic and primary lifestyle dimensions in bird niche hypervolume. According to their primary lifestyle, birds can be classified as aerial, insessorial, terrestrial, aquatic, or generalist. So, aerial birds forage in the air, terrestrials forage over the ground, and aquatics in the water. Generalist birds are those that can forage in various ways (on the ground, in the air, in trees). Insessorial category includes arboreal birds, and those that usually are perching on other substrates as buildings.

Between these categories based on habitat and locomotion modes, it exists morphological differences. However, it remains unclear if those groups are naturally congruent with the number of shared characteristics by the bird species included in them and likewise, at which extension we can use those traits to quantitatively predict the primary lifestyle of this group of animals. Because of that, and the importance of quantitative trait data, the objective of this work is to assess the pertinence of splitting birds according to the primary lifestyle classification that exists and, by doing so, the potential of morphological data to predict bird species primary lifestyle through machine learning.

## Materials and methods

### Data

For this study, I employed the Supplementary Dataset 1 of the AVONET project, which [4] released. This data includes almost all extant species of birds. I used the averaged traits at the species level according to the taxonomic treatment of BirdLife International (n = 11 009 species, 95.7–96.8% of extant bird species). From this dataset, I selected twelve variables: beak length measured from tip to skull along the culmen (BLC), beak length measured from the tip to the anterior edge of the nares (BLN), beak depth, beak width, tarsus length, wing length from carpal joint to wingtip measured on the unflattened wing, secondary length from carpal joint to tip of the outermost secondary (SL), Kipp's distance, and tail length, the corporal mass, the hand-wing index (HWI), and the primary lifestyle (eleven numeric variables and one categorical). As a previous step for further analysis, I skimmed the dataset to eliminate missing observations. For the subset of variables, I used there were no missing observations. All the statistical processing was developed in R 4.2.3 [20].

### Descriptive and univariate statistical analysis

As the first step of exploratory data analysis, I tested for the premises of normality for each quantitative variable. For this task, I used the function lillie.test from the R package nortest [21]. This function performs the Lilliefors (Kolmogorov-Smirnov) test according to [22]. To generate a visual representation of the relationship between observed data and expected distribution under normality, I constructed QQ plots with the function ggqqplot from the R package ggpubr [23]. Likewise, I explored the presence of skewness in the numeric data with the function skewness from the former R package.

I assessed the presence of differences between primary lifestyle for every one of the numeric variables. Univariate tests are the first approach to seek for morphological trends when taking in consideration ecological categories. For this procedure, I implemented an independence test based on permutations using a two-sided alternative hypothesis through the function independence_test of the R package coin [24, 25]. I used 10 000 Monte Carlo replicates to calculate the conditional null distribution. As *a posteriori* test, I ran a pairwise permutation test, using 0.05 as a threshold and the Benjamini & Hochberg correction [26] as a means of controlling family-wise error rate. In this case, I employed the pairwisePermutationTest function of the R package rcompanion [27] and 10 000 Monte Carlo replicates. Rain plots generated with the R packages ggplot2 [28] and ggdist [29] were used to represent the univariate tests previously described. For the rain plots, only the variables hand-wing index and the length of the wings were plotted without scale transformation. The remaining numeric variables were transformed using the decimal logarithm to facilitate visualization.

## Defining the classification models

The focus of this paper is to evaluate how good the primary lifestyle of birds is predicted according to morphological traits. In other words, how well supported are those ecological categories by divergent morphological features. To evaluate that idea I employed an extreme gradient boosting algorithm to do the classification task, which was implemented with the utilities included in the R package xgboost [30]. Among the machine learning procedures available, the one selected for this research offers a better development on a wide range of classification benchmarks than other available alternatives [19]. Tree-based algorithm have been widely used in ecology modeling with high performance and good results [31].

First, I split data into two sets, a training set (70%) and a testing set (30%) following the recommendations of [32]. I took advantage of the function initial_split from the R package rsample [33] for that operation. All the feature engineering processing steps were designed over the training set and applied to the testing set with the bake function from the R package recipes [34] to avoid data leakage. I implemented the steps of centering and scaling on all data set arrays to assure that all numeric variables had a comparable unit of measure [32] and because pre-processing of data is fundamental to reaching high performance in classification tasks [35]. These two data transformations improve the numerical stability of some computations inherent to machine learning algorithms [36].

For the classification problem at hand, I design three plausible xgboost models. For the first one, I wanted to train a xgboost model based on resuming variables to predict the primary lifestyle, in this case Principal Components. The result of a Principal Component Analysis is the reduction of dimensionality, and the generation of few variables that contemplate high levels of the variation inherent to the original data. This set of new variables has desirable properties for ecological studies, such as the total independence of their values in terms of covariance.

However, most machine learning algorithms have better performance when the number of features increases [37]. For that reason, I built two more xgboost models using the centered and scaled morphological traits for predicting the primary lifestyle of birds. The second xgboost model for this work was a conservative one, and I trimmed off variables with correlations higher than 0.9 (in the positive and negative sense). The function correlate of the R package corrr [38] served for calculating Pearson coefficients. The eliminated variables were SL, wing length, and beak width. Finally, for the third xgboost model I included all morphological variables, because the presence of highly correlated variables sometimes brings extra information for tree-based models.

## Modeling

The modeling pipeline was the same for all three xgboost models prepared (Fig 1). The selection of the best set of parameters for those was based on 10-fold cross-validation, using the misclassification error as metric of performance (the lower the value of the metric, the better model performance). To assure computational efficiency, I used 4,000 optimization rounds, (although few cases required more than 1,000 iterations to converge), and the optimization process was aborted after 100 rounds with no improvement in the misclassification error (measure against overfitting). The sampling process for the folds' generation was stratified according to the categories of the response variable (primary lifestyle). The function employed to do the cross-validation was xgb.cv. I used, then, the best array of parameters (Table 1) and the function xgboost to construct each final xgboost model through the training set, also with a maximum of 4000 iterations and an early stop if no improvement of 100. To ensure replicability, the seed for random procedures was set in 1998.

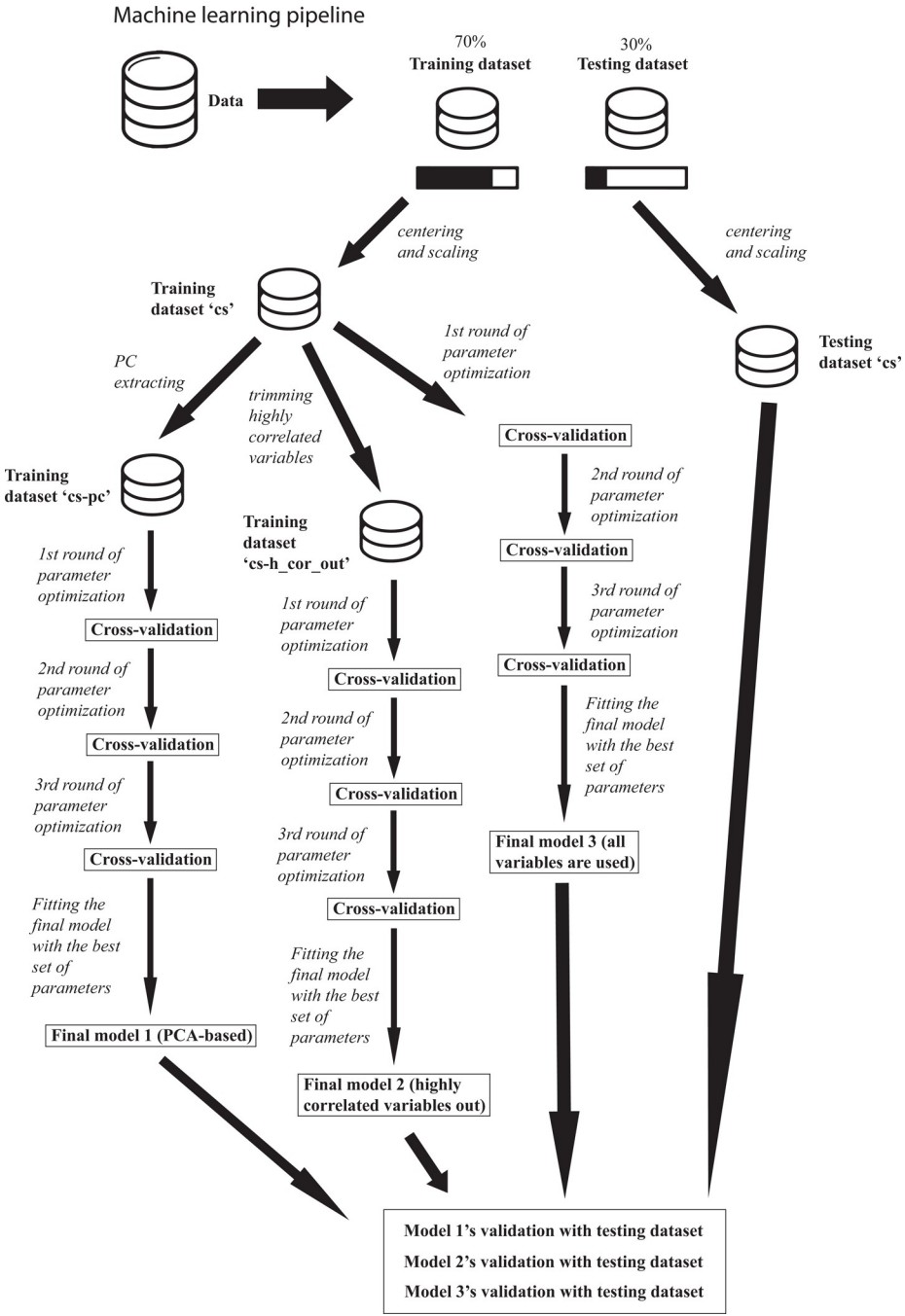

**Fig 1. Machine learning training pipeline.** The three extreme gradient boosting-based models were built through parameter optimization using the training dataset and validation with unforeseen observations, stored in the testing dataset. The best combination of parameters was extracted through 10-fold cross-validation. 'cs': centered and scaled dataset. 'cs-pc': centered and scaled training dataset with extraction of principal components. 'cs-h_cor_ou': centered and scaled training dataset leaving highly correlated variables out.

The importance of the variables was obtained through the function xgb.importance. Lastly, I validated the xgboost models against the corresponding testing data set. I generated the confusion matrix and performance statistic associated with the xgboost model validations through the function confusionMatrix of the R package caret [39].

**Table 1. Parameters used to train the xgboost final models through the extreme gradient boosting algorithm in this work (n = 11,009 species).**

| Xgboost Model | Classes | Booster | eta | M_D | M_C_W | α | ϒ | λ |
|---|---|---|---|---|---|---|---|---|
| 1 | 5 | gbtree | 0.1 | 10 | 2.0 | 1 | 0.1 | 1 |
| 2 | 5 | gbtree | 0.2 | 10 | 0.5 | 0 | 0.01 | 1 |
| 3 | 5 | gbtree | 0.2 | 8 | 0,5 | 0 | 0 | 0 |

The first xgboost model uses principal components from one to five, the second uses all variables but the highly correlated ones, and the third exploits every single variable. In all cases, morphological traits values were centered and scaled. eta: Learning rate. M_D: Maximum depth. M_C_W: Minimum child weight. α, ϒ, λ: Regularization parameters.

## Contrasting the best machine learning model with traditional approaches

For contrasting the best xgboost model obtained through the extreme gradient boosting algorithm, I implemented two classical approaches used for classification tasks. For that, I fit a multinomial logistic regression-based model using decay values between 0 and 0.00001 and selecting the best performance through the AIC criterium. For doing so, I worked too with the predictors scaled and centered, and I built the model with the function multinom from the R package nnet [40]. I used the testing split for assessing the performance of the model and the confusionMatrix function to obtain comparable results. Next, I ran a Linear Discriminant Analysis using the function lda from the MASS R package [40], which results were also evaluated through the function confusionMatrix.

## Results

### Descriptive and univariate statistical analysis

Every one of the numeric variables used in this study showed deviation from normality. The results of the normality test applied, and the Q-Q plots generated are available in S1 Fig. Moreover, every numeric variable distribution shows a positive skew, and the values of this measure range from 1.21 in the case of the Hand-Wing Index to 42.02 in the case of the animal mass.

The effect of the variable primary lifestyle was statistically significant when the categories were compared according to every numeric predictor (Fig 2). In the case of the BLC, BLN, beak width, beak depth, wing length, Kipp's distance, SL, and mass (Fig 2A–2D, 2F–2H and 2K), generally the highest values were present in the group defined by an aquatic primary lifestyle. The minimal values measured for BLC, BLN, beak width, beak depth, tarsus length, SL, and mass were concentrated in the aerial birds (Fig 2A–2E, 2H and 2K). The category with the major hand-wing index, globally speaking, was the aerial one, followed by the aquatic birds (Fig 2I).

All pairwise comparisons according to the beak width, SL variables, and mass were statistically significant. In the case of the variables BLN, beak depth, tarsus length, wing length, Kipp's distance, hand-wing index, only a pairwise comparison resulted statistically non-significant (Fig 2B, 2D–2G and 2I). For BLC, the pairs aerial-generalist and generalist-insessorial were not differentiable according to the *a posteriori* test. The most homogeneous (in this case, because of the existence of more than two similar pairs) variable was the tail length, and for this variable, it did not exist differences between the pairs aerial-insessorial, aquatic-generalist, aquatic-terrestrial, and generalist-terrestrial.

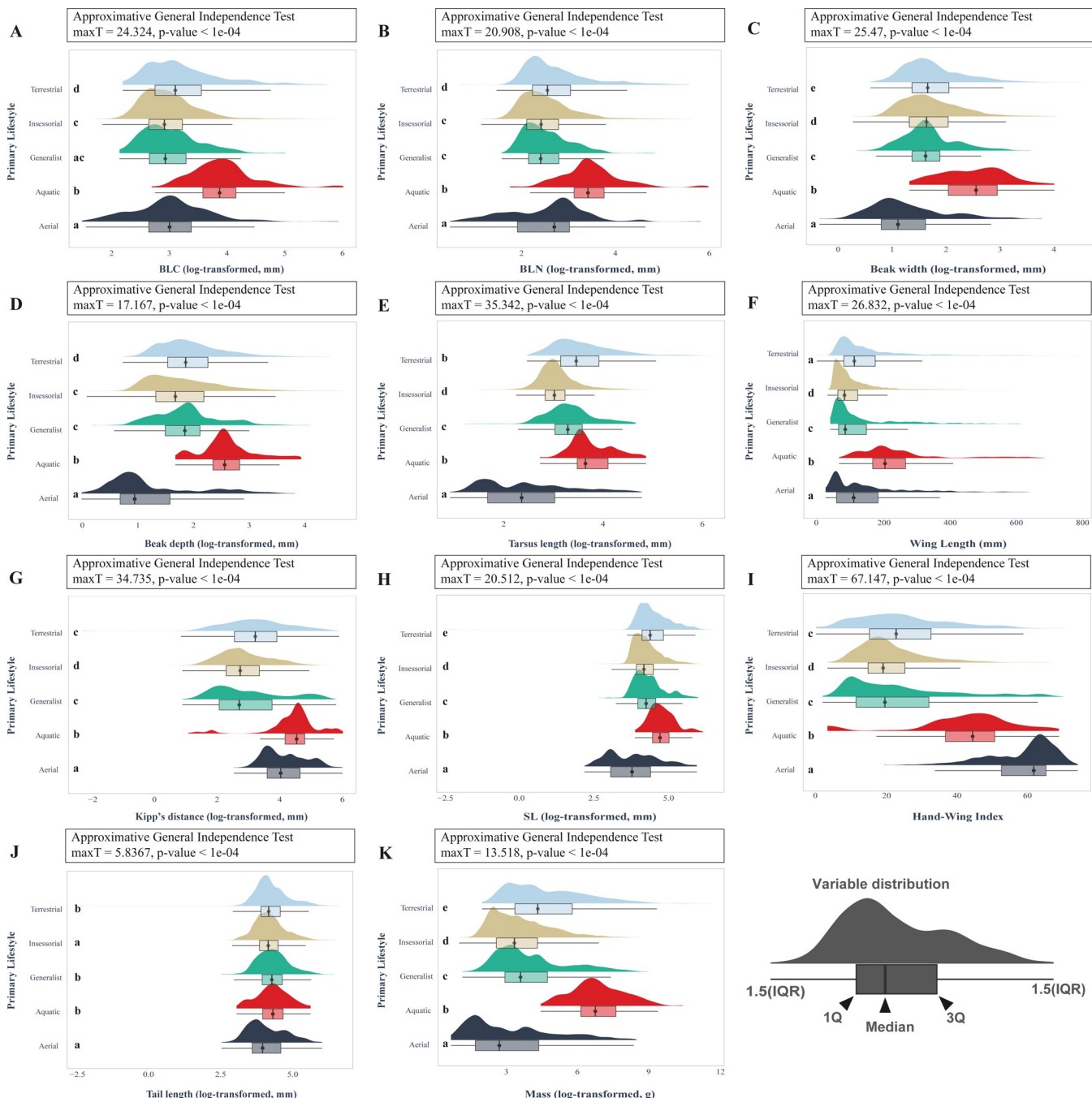

**Fig 2. Rain plots for every predictor used to visually check the existence of differences between primary lifestyle in birds (n = 11 009 species).** In the boxplot, whiskers represent 1.5 times the interquartile range, the boxes edges are the first and third quartiles, and the central region is the median. Each plot has the corresponding results of the approximative independent test based on Monte Carlo replicates. Every different letter is used for denotate distinctive groups according to the *a posteriori* pairwise permutation test. A. Beak length measured from tip to skull along the culmen (BLC). B. Beak length measured from the tip to the anterior edge of the nares (BLN). C. Beak width. D. Beak depth. E. Tarsus Length. F. Wing length from carpal joint to wingtip measured on the unflatten wing. G. Kipp's distance H. Secondary length from carpal joint to tip of the outermost secondary (SL). I. Hand-Wing Index J. Tail length. K. Mass. Expect wing length and Hand-Wing index, the other variables were log transformed.

### Defining the classification xgboost models and modeling

The PCA showed that the first five principal components, which summarize 94% of the data variance, do not resolve effectively the five classes of birds according to the primary lifestyle (Fig 3). All five classes have an extensive overlapping between each other according to the ellipses obtained using the principal component scores. However, there exists a subtle segregation if every component is analyzed individually. In the PC1, density plot shows a separation between aquatic birds and the rest of categories, which tends to cluster together. In the case of PC2, aerial individuals have lower score values, while terrestrial and insessorial ones have higher values.

The Pearson-based correlation analysis shows that five pairs of variables had a coefficient of correlation higher than 0.9, umbral selected in this study to trim strongly related measures. The pairs were BLN-BLC, beak depth-beak width, SL-wing length, mass-wing length, and mass-SL (Fig 4). Around these variable pairs were performed the variable trimming process described in materials and methods to prepare the data for constructing xgboost model 2. The rest of the variable's pairs showed correlation coefficients below 0.9.

### Modeling

All three xgboost models built showed high levels of accuracy (78.9% for xgboost model 1, 83.1% for xgboost model 2, and 84.1% for xgboost model 3). The confidence intervals at 95% for these measures are (0.775, 0.8031), (0.8173, 0.8432), and (0.8276, 0.8528), respectively. In all cases, Cohen's kappa value was superior to 0.6, being the maximum value achieved at 0.72 (xgboost model 3) and the minimum at 0.62 (xgboost model 1). In all cases, the No Information Rate was statistically significantly lesser than the accuracies of the models, with a probability value associated in the three models minor than 2.2e-16.

Regarding the results of performance by class (Table 2), the three xgboost models showed the highest sensitivity or recall for class insessorial and the lower values for generalist birds. On the other side, the xgboost models were highly specific for aerial, aquatic, and generalist ones, in all cases with values of specificity above or equal to 0.97. The precision or positive predictive value (PPV) ranged from 0.50 to 0.85 for xgboost model 1, from 0.58 to 0.87 for xgboost model 2, and 0.53 to 0.89 for xgboost model 3. In all cases, PPV was higher for classes aerial and insessorial birds. Negative predicted values were superior to 0.86 in all models, peaking for aerial and aquatic categories. The balanced accuracy of xgboost model 3 was the highest for generalist, insessorial, and terrestrial birds of the three xgboost models built. However, xgboost model 2 was the best delimitating the aerial class. To summarize, sensitivity, specificity, PPV, NPV, and F1 measures were almost always greater for xgboost model 3 when compared to the other two models (the exception is related to the aerial class, which was best described by xgboost model 2).

To comprehend better the previous results, I constructed a confusion matrix for each xgboost model (Fig 5), with the by-class sensibility reflected in the diagonal line. As can be appreciated, almost all cases are concentrated in the diagonal line, which corresponds to the proportion of correctly classified primary lifestyles. Xgboost model 2, which occupies the second place in overall performance, had the highest values for sensitivity for class aerial. The sensitivity for the class generalist in xgboost model 3 was higher than in previous models, the same is also true for classes terrestrial and insessorial. However, each model had troubles in the classification task. Xgboost models 1 and 2 misclassified extensively generalist birds, assigning them to the classes insessorial and terrestrial in great proportion, even major that its classification as generalist. It also put frequently some terrestrial organisms in the category of insessorial, and aquatic birds in the class terrestrial (in this description, I only talked about

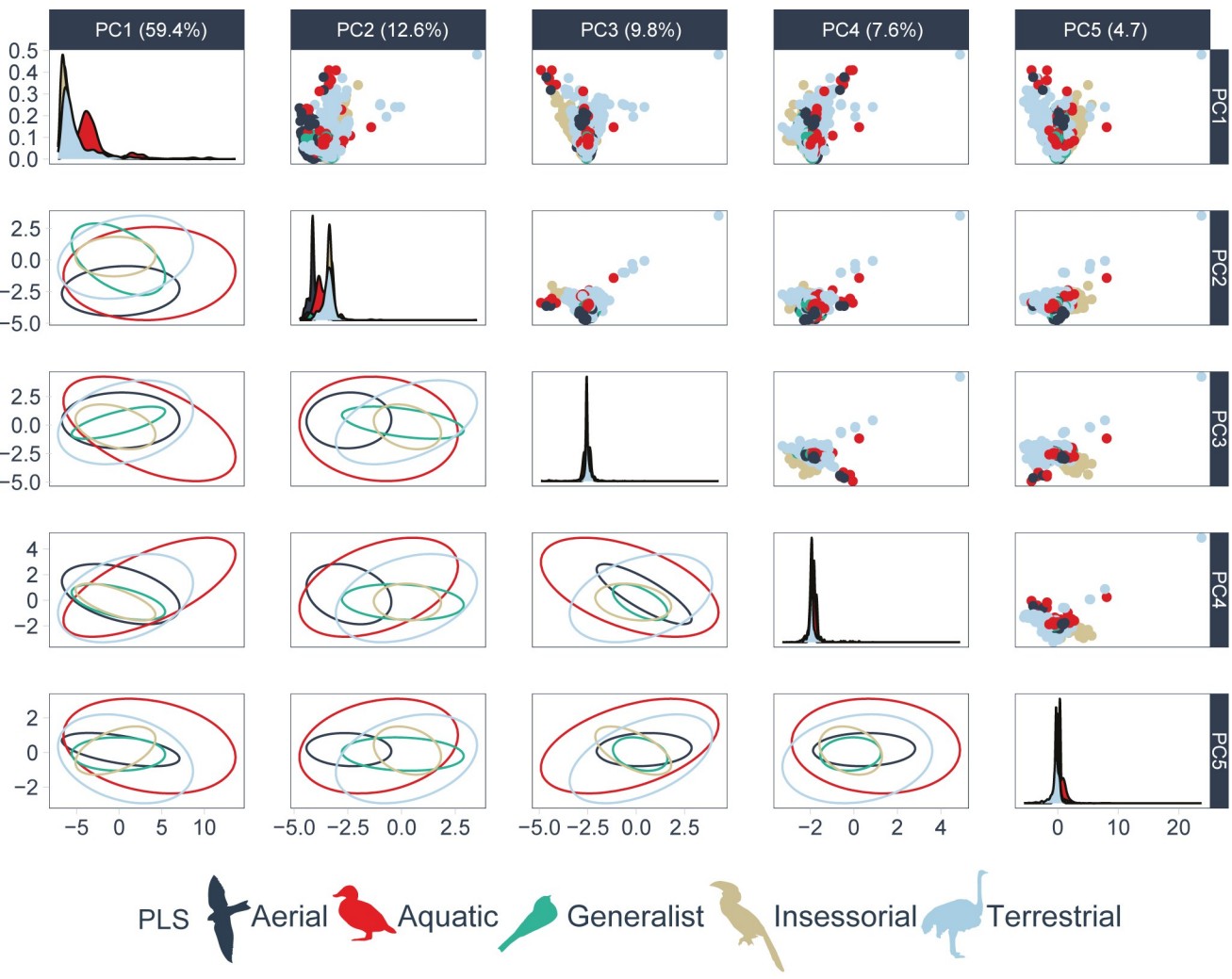

**Fig 3. Bi-dimensional representation of the bird primary lifestyle through the pairwise combination of components derived from a principal components analysis (n = 11 009 species).** The dimensionality reduction was made after data centering and scaling.

misclassification with proportions higher than 0.1). Xgboost model 3 showed the best performance, yet it also misclassified generalist birds as insessorial and terrestrial.

The importance of the predictors is reflected in Fig 6. For xgboost model 1, PC2, PC5, and PC4 were the most influential variables in the classification task (in that same order). For xgboost model 2 and xgboost model 3, the Hand-Wing index, tarsus length, and mass were the ones with higher relevance for the classification task. Note how the order of the shared variables by models two and three is the same according to importance.

## Contrasting the best machine learning model with traditional approaches

Although the models built using multinomial logistic regression and linear discriminant analysis were statistically significantly higher in accuracy predicting the primary lifestyle relative to the no information rate (p < 2.2e-16), their overall accuracies were poor relative to the three xgboost models built through extreme gradient boosting (0.74 and 0.70, respectively). Moreover, the ability to correctly recognize generalist birds was almost zero (Table 2). Kappa values

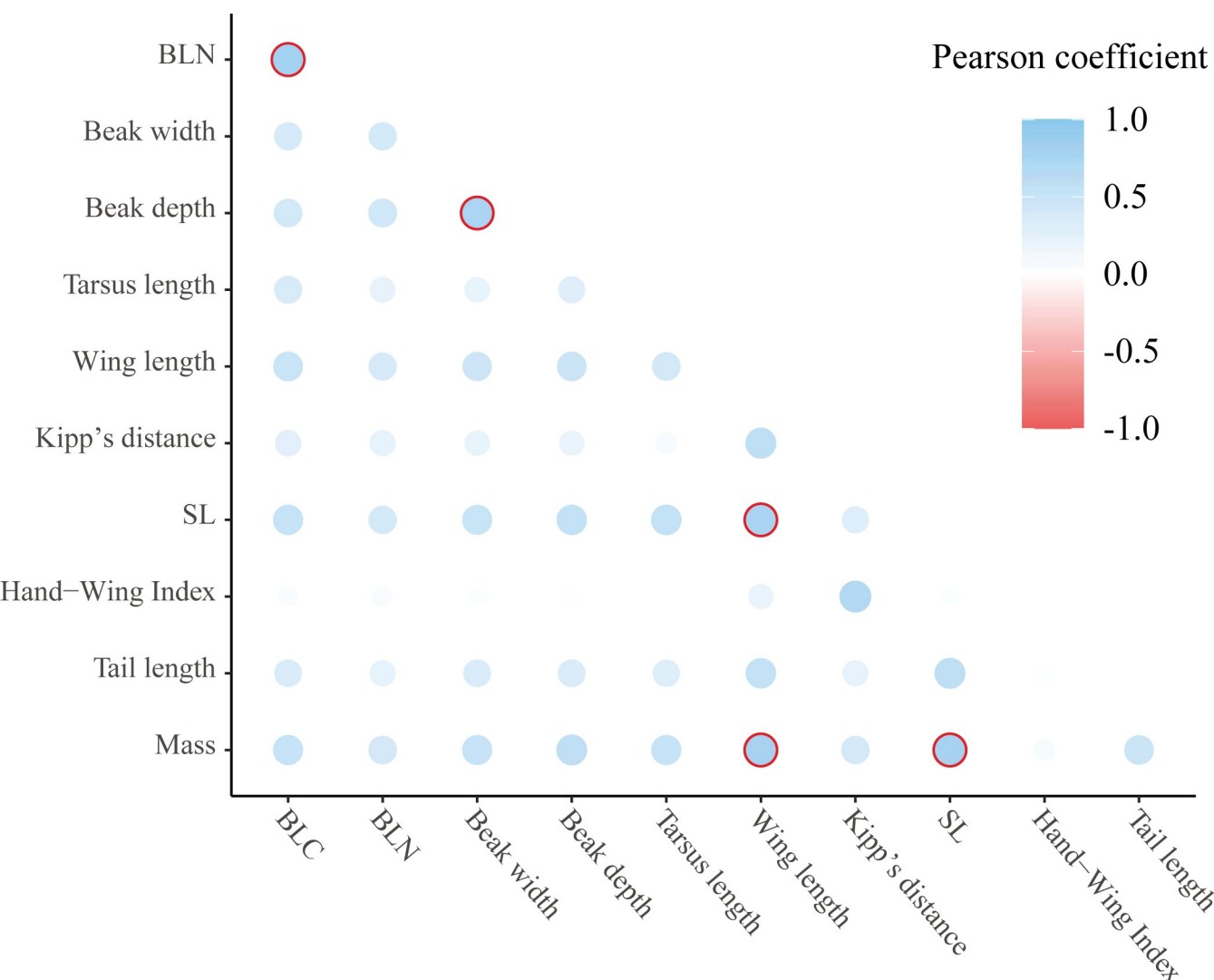

**Fig 4. Pearson correlation coefficients obtained for each variable pair (n = 11 009 species).** The blue color is related with positive values, while red color represents negative correlation (not evinced in this study). Encircled represents all the coefficients that are superior to the threshold declared in materials and methods (0.9). By trimming one of the variables of every encircled pair, the data set was prepared for constructing xgboost model 2.

were also substantially lesser, with 0.50 for the multinomial logistic regression and 0.39 for the linear discriminant analysis.

## Discussion

In this research, a set of eleven morphological traits was employed to link functional traits and primary lifestyles in birds. All variables were not normally distributed and showed a positive skewness. However, skewness was not a problem in this investigation because tree-based models are insensitive to this feature problem [32]. Generally, there were significant differences between most primary lifestyles of birds using univariate comparisons. These comparisons showed some trends of ecological relevance. For instance, aquatic birds generally have greater mass, wing length, beak depth, and width, BLN, and BLC. Moreover, aerial birds evince, as category, major values of the Hand-Wing Index. The hang-wing index is a derived measure that reflects flight efficiency and the dispersion ability of birds [4, 41]. So, it is logical to expect that

**Table 2. Performance measures for every model built to predict bird primary lifestyle through morphological traits (n = 11 009 species).** The first three models were built using extreme gradient boosting algorithm (called xgboost models to avoid confusions), and the last two correspond to traditional approaches using the same features (all morphological traits) than the best xgboost model trained through extreme gradient bosting: multinomial logistic regression (MLR) and linear discriminant analysis (LDA). The last two models offer a contrast between traditional approaches used in classification tasks and the machine learning algorithm used in this research in front of dimensionality of features.

| Primary Lifestyle | Sens | Spec | PPV | NPV | Prec | F1 | Prev | D. Rate | D. Prev | B. Accur |
|---|---|---|---|---|---|---|---|---|---|---|
| Xgboost Model 1 (PC1 - PC5) **Overall accuracy: 78.93%** | | | | | | | | | | |
| Aerial | 0,89 | 0,99 | 0,85 | 0,99 | 0,85 | 0,87 | 0,07 | 0,07 | 0,08 | 0,94 |
| Aquatic | 0,78 | 0,99 | 0,74 | 0,99 | 0,74 | 0,76 | 0,03 | 0,02 | 0,03 | 0,89 |
| Generalist | 0,26 | 0,97 | 0,50 | 0,93 | 0,50 | 0,34 | 0,09 | 0,02 | 0,05 | 0,62 |
| Insessorial | 0,92 | 0,72 | 0,83 | 0,86 | 0,83 | 0,87 | 0,59 | 0,55 | 0,66 | 0,82 |
| Terrestrial | 0,61 | 0,93 | 0,70 | 0,90 | 0,70 | 0,66 | 0,22 | 0,13 | 0,19 | 0,77 |
| Xgboost Model 2 (highly correlated variables out) **Overall accuracy: 83.05%** | | | | | | | | | | |
| Aerial | 0,91 | 0,99 | 0,87 | 0,99 | 0,87 | 0,89 | 0,07 | 0,07 | 0,08 | 0,95 |
| Aquatic | 0,83 | 1,00 | 0,86 | 1,00 | 0,86 | 0,85 | 0,03 | 0,02 | 0,02 | 0,91 |
| Generalist | 0,34 | 0,98 | 0,58 | 0,94 | 0,58 | 0,43 | 0,09 | 0,03 | 0,05 | 0,66 |
| Insessorial | 0,94 | 0,77 | 0,86 | 0,90 | 0,86 | 0,90 | 0,59 | 0,56 | 0,65 | 0,86 |
| Terrestrial | 0,70 | 0,95 | 0,78 | 0,92 | 0,78 | 0,73 | 0,22 | 0,15 | 0,19 | 0,82 |
| Xgboost Model 3 (all variables) **Overall accuracy: 84.05%** | | | | | | | | | | |
| Aerial | 0,88 | 0,99 | 0,89 | 0,99 | 0,89 | 0,88 | 0,07 | 0,07 | 0,07 | 0,94 |
| Aquatic | 0,84 | 1,00 | 0,84 | 1,00 | 0,84 | 0,84 | 0,03 | 0,02 | 0,03 | 0,92 |
| Generalist | 0,35 | 0,97 | 0,53 | 0,94 | 0,53 | 0,42 | 0,09 | 0,03 | 0,06 | 0,66 |
| Insessorial | 0,94 | 0,81 | 0,88 | 0,90 | 0,88 | 0,91 | 0,59 | 0,56 | 0,63 | 0,88 |
| Terrestrial | 0,76 | 0,94 | 0,79 | 0,93 | 0,79 | 0,77 | 0,22 | 0,16 | 0,21 | 0,85 |
| MLR (all variables) **Overall accuracy: 74.49%** | | | | | | | | | | |
| Aerial | 0,84 | 0,98 | 0,80 | 0,99 | 0,80 | 0,82 | 0,07 | 0,06 | 0,08 | 0,91 |
| Aquatic | 0,39 | 0,99 | 0,56 | 0,98 | 0,56 | 0,46 | 0,03 | 0,01 | 0,02 | 0,69 |
| Generalist | 0,04 | 0,99 | 0,42 | 0,91 | 0,42 | 0,08 | 0,09 | 0,00 | 0,01 | 0,52 |
| Insessorial | 0,97 | 0,54 | 0,76 | 0,92 | 0,76 | 0,85 | 0,59 | 0,58 | 0,76 | 0,76 |
| Terrestrial | 0,43 | 0,95 | 0,69 | 0,86 | 0,69 | 0,53 | 0,22 | 0,09 | 0,13 | 0,69 |
| LDA (all variables) **Overall accuracy: 69.85%** | | | | | | | | | | |
| Aerial | 0,85 | 0,97 | 0,72 | 0,99 | 0,72 | 0,78 | 0,07 | 0,06 | 0,09 | 0,91 |
| Aquatic | 0,40 | 0,98 | 0,31 | 0,98 | 0,31 | 0,35 | 0,03 | 0,01 | 0,03 | 0,69 |
| Generalist | 0,00 | 1,00 | 0,50 | 0,91 | 0,50 | 0,01 | 0,09 | 0,00 | 0,00 | 0,50 |
| Insessorial | 0,98 | 0,42 | 0,71 | 0,93 | 0,71 | 0,83 | 0,59 | 0,58 | 0,81 | 0,70 |
| Terrestrial | 0,20 | 0,97 | 0,67 | 0,82 | 0,67 | 0,31 | 0,22 | 0,04 | 0,06 | 0,59 |

**Sens**: Sensibility or recall. **Spec**: Specificity. **PPV**: Positive predictive value. **NPV**: Negative predictive value. **Prec**: Precision. **Prev**: Prevalence. **D. Rate**: Detection rate. **D. Prev**: Detection prevalence. **B. Accur**: Balanced Accuracy.

birds that spend much of the time in flight and hunt or forage predominantly on the flight, have greater flight efficiency.

Based on the results obtained in this work, it seems that all three extreme gradient boosting models were successful in linking morphological traits to primary lifestyle in birds. Xgboost model 3 showed the best performance, with an accuracy of 84.05% and a Cohen's kappa value of 0.72, indicating a considerable level of agreement between the actual data and the model's predictions. The high accuracy and Cohen's kappa values for all three xgboost models reflect that morphological traits are indeed useful predictors of primary lifestyle in birds.

The xgboost models were particularly successful in predicting the lifestyles of insessorial and aerial birds, with high sensitivity and specificity for these classes. Xgboost model 3 was

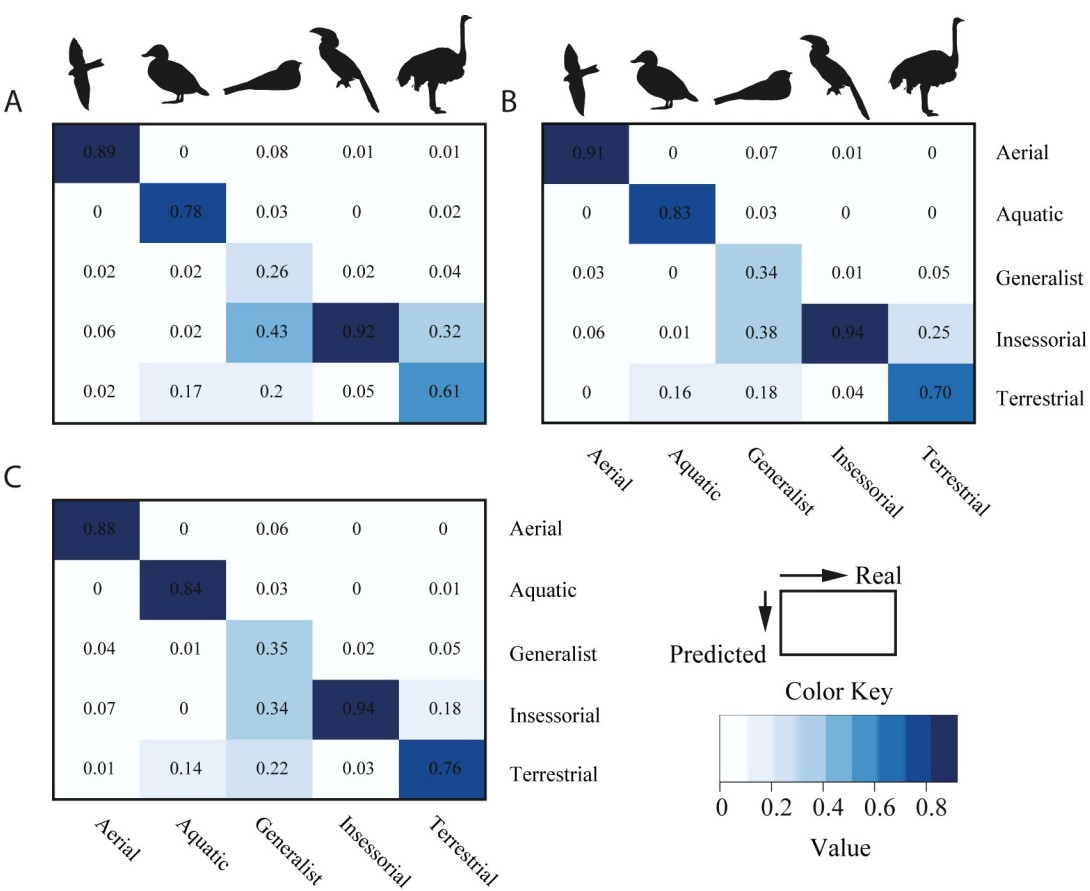

**Fig 5. Confusion matrix for every xgboost model build to predict bird primary lifestyle through morphological traits (n = 11 099 species).** The more intense the blue color, the higher the proportion in a particular cell. The diagonal line represents the sensitivity or recall values. The columns have the real categories, while the row presents the predicted ones. A. Xgboost model 1, which was built using only the first five principal components. B. Xgboost model 2, which was built leaving out highly correlated variables. C. Xgboost model 3, which was built using all variables. Note: As a previous step for modeling, all numeric variables were scaled and centered.

able to predict aquatic and terrestrial primary lifestyles also with good performance. The xgboost models had lower sensitivity for generalist birds, which may suggest that generalist birds have more diverse morphological traits that are harder to predict. However, the xgboost models were still highly specific for generalist birds, indicating that they were successful in correctly identifying birds that were not generalists.

An accuracy value in the range of 0.75 to 0.85 indicates that the xgboost models provide a considerable improvement over random guessing in the classification task of this work. One potential source of error in this kind of algorithm is the quality and quantity of data employed if data is incomplete or biased. Nevertheless, this is not the case with this research, which used information from almost every single species of bird. Complexity does not seem to be a problem either and the most complex xgboost model was the one with the greater performance.

In works that attempt to link functional traits to function, according to [7], a key question is whether species distribution in a multidimensional space is related to ecological function. In their work, they tested combinations of morphological traits to predict trophic niches and broad trophic levels. They found that using only mass, predictive power in either classification task did not overcome 40%. Next, they included beak size and shape, and the accuracy

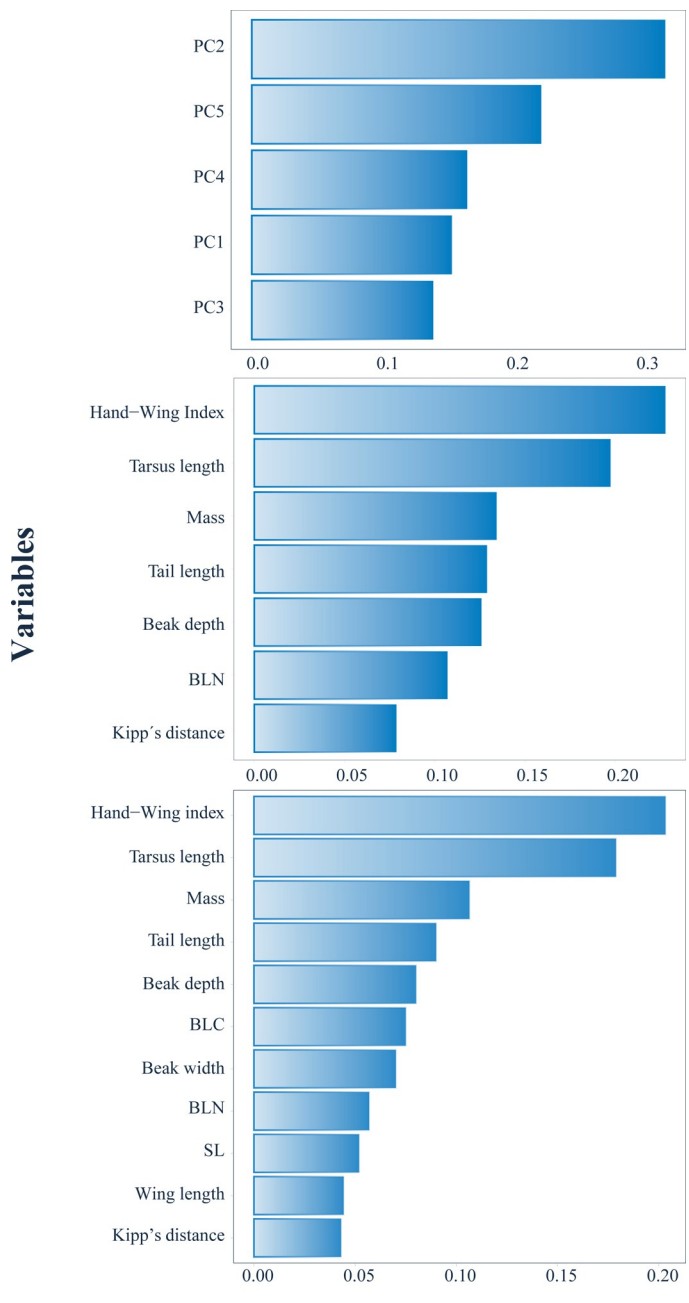

**Fig 6. Importance of predictors used to build the xgboost models of classification for identifying bird primary lifestyle through morphological traits (n = 11 009 species).** A. Xgboost model 1, which was built using only the first five principal components. B. Xgboost model 2, which was built leaving out highly correlated variables. C. Xgboost model 3, which was built using all variables. Note: As a previous step for modeling, all numeric variables were scaled and centered.

increased to 78%. The use of nine morphological traits improved the predictability above 80%, a situation like the one in this work, where the most complex model classified the birds with higher accuracy. These authors found that each trophic niche and trophic level occupies a

distinctive location of the bird morphospace. The xgboost models of this work were reasonably similar in performance to the best model obtained by [7], although, these authors were predicting through random forests trophic niche, not primary lifestyle.

As in the study conducted by [7], machine learning algorithms seems to be the best choice in front of complex data. In this work, classical approaches used for predicting categorical variables -as the primary lifestyle- through continuous predictors -as morphological traits-, performed poorly in comparison with machine learning based ones, even if it is taken as reference xgboost model 1 (Principal components-based one). These elements highlight the advantages of machine learning techniques to deal with complex and highly dimensional data. There is a set of reasons why classical approaches are not good choices for dealing with complex data, and always are related to the called "dimensionality curse" [37]. For instance, in the case of multinomial logistic regressions, they may suffer from multicollinearity, overfitting, or separation issues if the predictors are highly correlated or have extreme values. All these conditions are evinced in the data for this work.

The dimensional reduction procedure executed in this investigation shows how the relationship between morphological traits and ecological categories is extremely complex. A bidimensional representation based on the PCA decomposition and a lineal space was unable to delineate the primary lifestyle of birds. This result also shows the advantages of machine learning algorithms to detect high-level and other relations than linear between variables. Machine learning algorithms, because of their ability to learn from data without making strong assumptions, are powerful tools in classification tasks [32].

Regarding variable importance, it is necessary to analyze the relevance of variables in xgboost models 2 and 3 (xgboost model 1 is based on principal components). In xgboost model 2 and 3, the most important variables were the Hand-Wing index, tarsus length, and mass. Morphological differences in variables associated with wings, tails, mass, and legs are related to locomotion, and because of that, they offer insights into how birds move in their environment and search for food or resources [4]. Likewise, variables more aligned with trophic categories–as beak features–were less important in the models of this work. However, it is noticeable that mass occupies position three in this hierarchy, even though previous studies, like the one conducted by [7], have found that mass offers a relatively weak explanation for some ecological factors - such as trophic niche. Also, contrary to what one could expect, wing length, also associated with flight activity, occupies the penultimate position in the importance chain in xgboost model 3. From this result, it could be concluded that for primary lifestyle, wing length is not relevant if it is not taken in relation to the Kipp's distance (that ratio is the Hand-Wing index). That relation is the one that provide a relative measure of efficiency in the flight activities [4], which must be the most important element associated to primary lifestyles in birds according to xgboost model 3.

Misclassification could be derived from the fact that primary lifestyle categories include birds with overlapping morphological features. This is an aligned factor related to other ecological variables, such as the trophic niche. For instance, fish-eating species might be aerial, aquatic, insessorial, or terrestrial [4]. This could be considered a potential source of uncertainty in the classification process executed in this research. Also, is important to mention that foraging niches in the study of [7] were less predictable in comparison with trophic niches. Furthermore, the identity and amount of dimensions required to predict avian niches accurately are niche-specific, and this element reflects the contrasting diverse life modes and strategies of birds [7]. Of course, this is a limitation for model training, because normally the set of dimensions selected for classification tasks is applied globally. This could be the cause of the misclassification of certain classes of birds in this kind of research.

Another element to consider in the evaluation of models linking morphological traits and ecological features is that there exists a wide range of morphological traits that are multifunctional and, because of that, understanding their evolution under the scope of adaptation and constraints is, at least, challenging. Under this premise, even when a trait is essential to a particular function, trait divergence among species could be directed by a second and lesser obvious function [11]. Under this idea, despite the precept that beak morphology represents the evolutionary record of adaptation to trophic niches, evidence also promotes the idea that beak morphology and diversity could be shaped by other factors, such as phylogeny and other morphological traits. So beak cannot evolve as an independent unit [10]. Then, it is pertinent to assess in future studies the multifunctional nature of the traits used to build the models, to control other aligned factors that also shape the morphology of birds.

It is important to notice that primary lifestyle as an ecological guild for birds could be inherently complex. According to [42], the actual guild in nature might be a multilevel classification, which means that the structure could be hierarchical. In this sense, if we indeed take the primary lifestyle of birds as a guild, there could be categories within as function of other ecological variables.

Finally, as showed in this work and in others related, birds are suitable organisms to perform trait-based studies with wide coverage. First, the clade of the birds is lower in number than plants, and this makes their study more achievable in terms of morphological features coverage. Second, the worldwide distribution of birds across oceans and biomes is a desirable property to assess effectively morphoecological links. Finally, they are the best studied on a global scale [3]. Despite these advantages, this clade also has a flaw in modeling tasks, its huge ecomorphological variance in the context of bird species evolutive radiation [7].

## Conclusions

Birds are a diverse group of vertebrates that occupy a wide range of ecological niches and display a variety of morphological adaptations. Morphological traits, such as body size, bill shape, wing shape, and leg length, can affect the ability of a bird to exploit different food resources and habitats, as well as to avoid predators and competitors. Therefore, there is a close relationship between morphology and ecology in birds, which can be studied at different levels of biological organization: from individuals to populations, communities, and phylogenies. Overall, the results of this research provide valuable insights into the relationship between morphological traits and primary lifestyle in birds and demonstrate the potential for using machine learning models to accurately predict bird lifestyles based on their physical characteristics. First, using appropriate models, we can establish a link between the morphology and primary lifestyle of bird species. This suggests that, generally speaking, primary lifestyles have a substantial morphological basis. Second, it could be pertinent to review current division of birds according to primary lifestyle, maybe using this division in combination with other ecological grouping variables to stablish a hierarchical division in more comprehensive guilds, given that there were problems in the delimitation of the generalist category. That could improve the accuracy of models by providing more natural ways of grouping birds according to shared characteristics. Third, machine learning algorithms could be exploited to assess the relevance of features while clustering organisms if there exists an extended morphological variability and correlation between traits. Finally, misclassifications in the models fitted for this work show that primary lifestyles are not exclusive categories, and they evince overlapping in the bird morphospace.

## Data and code availability

Data is fully available. See [4]. R script is available in a public repository of GitHub. See https://github.com/luismadrigal98/Predicting-bird-lifestyle-with-morphology-and-machine-learning.

## Supporting information

**S1 Fig. QQ plots for every predictor used to assess the relation between morphological traits and the primary lifestyle of birds.** Each plot has the corresponding results of the Lilliefors (Kolmogorov-Smirnov) test of normality, which are the statistic D and the probability value associated. A. Beak length measured from tip to skull along the culmen (BLC). B. Beak length measured from the tip to the anterior edge of the nares (BLN). C. Beak width. D. Beak depth. E. Tarsus Length. F. Wing length from carpal joint to wingtip measured on the unflattened wing. G. Kipp's distance H. Secondary length from carpal joint to tip of the outermost secondary (SL). I. Hand-Wing index J. Tail length. K. Mass.
(TIF)

## Acknowledgments

I would like to express my gratitude to my colleagues, friends and family for their unwavering support and encouragement throughout this research journey, and specially to Dr. John K. Kelly, my current advisor, who has supported me in these last times for publishing this work. I would also like to thank the reviewers for their valuable feedback and suggestions that helped improve the quality of this paper.

## Author Contributions

**Conceptualization:** Luis Javier Madrigal-Roca.

**Data curation:** Luis Javier Madrigal-Roca.

**Formal analysis:** Luis Javier Madrigal-Roca.

**Investigation:** Luis Javier Madrigal-Roca.

**Methodology:** Luis Javier Madrigal-Roca.

**Software:** Luis Javier Madrigal-Roca.

**Supervision:** Luis Javier Madrigal-Roca.

**Validation:** Luis Javier Madrigal-Roca.

**Visualization:** Luis Javier Madrigal-Roca.

**Writing – original draft:** Luis Javier Madrigal-Roca.

**Writing – review & editing:** Luis Javier Madrigal-Roca.

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
