## [Decision Letter · Decision Letter 0]

30 Aug 2023

PONE-D-23-14687Assessing the predictive value of morphological traits on primary lifestyle of birds through the extreme gradient boosting algorithmPLOS ONE

Dear Dr. Madrigal-Roca,

Thank you for submitting your manuscript to PLOS ONE. After careful consideration, we feel that it has merit but does not fully meet PLOS ONE’s publication criteria as it currently stands. Therefore, we invite you to submit a revised version of the manuscript that addresses the points raised during the review process.

We look forward to receiving your revised manuscript.

Kind regards,

Vitor Hugo Rodrigues Paiva, Ph.D.

Academic Editor

PLOS ONE

Journal Requirements:

Reviewers' comments:

Reviewer's Responses to Questions

**Comments to the Author**

1. Is the manuscript technically sound, and do the data support the conclusions?

Reviewer #1: Partly

Reviewer #2: Partly

2. Has the statistical analysis been performed appropriately and rigorously? 

Reviewer #1: I Don't Know

Reviewer #2: I Don't Know

3. Have the authors made all data underlying the findings in their manuscript fully available?

Reviewer #1: Yes

Reviewer #2: Yes

4. Is the manuscript presented in an intelligible fashion and written in standard English?

Reviewer #1: No

Reviewer #2: Yes

5. Review Comments to the Author

Reviewer #1: Throughout I think the objective of this research is not emphasised strongly but particularly in the abstract and the opening introduction paragraph.

The abstract states that "This paper aims to evaluate the predictive potential of morphological data to determine the primary lifestyle of birds through a machine learning algorithm." but not why you would want to do that, or how it will help solve various ecological questions. Is the aim just to show that you can? It isn't very surprising that you can predict at some level primary lifestyle from morphology. Or are you trying to show a good method for doing so? What is the measure of being able to determine lifestyle from morphology? With modern machine learning you can build some level of prediction for most things given enough data.

The abstract goes into too much methodology.

"The best model generated, for which all 11 morphological traits were used, had an accuracy of 83%.": I don't feel that this is a very good prediction level for something as basic as primary lifestyle given everything has been included. What does a model such as just wing span and beak length/width give you? Of these 11 traits, many are going to be highly correlated, for example body mass has continually been shown to be correlated to wing-span and beak size. Is this model over-fitted? It would be good to see some analysis of number of traits to prediction accuracy - possibly plotted as (x number of traits, y prediction power) for both the test and training set. It would also be good to see some analysis for tuning nrounds/eta and depth (possibly also colsample_bytree,subsample and min_child_weight - from stackoverflow articiles on reducing overfitting).

The introduction needs rework to include the motivation behind this work. Also a better focus on the aim, for prediction what would be an acceptable prediction accuracy (based on what previous work?). What are the trade offs for including more traits vs a simpler model?

In the methods it is not clear to me modelling approach has been chosen. What other approaches were considered? Why were they discarded? It is a very step by step procedure of what was done, but I don't really understand why those decisions were made, what trade offs were involved, or the reasoning for the steps. I find it rather technical and difficult to follow, and don't reach the end understanding why choices have been made. Particularly, why only 3 models with these parameters?

In the results I am not sure that comparing each morphological trait to primary lifestyle is particular interesting or related to the aims of this paper? I think a better stated aim would improve the results. For figure 2 it would be good to include all PCA axis, and possibly also transform the PCA (e.g. log) to make the plot clearer. Table 2 and figure 4 do not add anything. I do not understand the significance of figure 5.

In the conclusion it would be good to know how this paper will allow us to better answer ecology questions about birds.

"Overall, the results of this research provide valuable insights into the relationship between morphological traits and primary lifestyle in birds and demonstrate the potential for using machine learning models to accurately predict avian lifestyles based on their physical characteristics." - what were the valuable insights?

In general, the paper is very technical with a lot of technical jargin that I found hard to follow and seems to go very deep into how these 3 models were built. But I don't think I really understanding the reasoning for building these three models. I do think there are some interesting questions to answer here and a good paper can be achieved with some better framing, more hand-holding through the methods, and tying the whole paper together with a stronger aim.

Reviewer #2: Please see attached file. I copy below only the general comments but all details are in the attached pdf.

I found the topic of your research very interesting but also encountered important structural problems. I did not comment on the entirety of the manuscript as I think that most of my comments apply to all the sections in the text. In general I would identify the main weakness of this work in the structure of the text, which is often convoluted and hides the main messages/results of the study. A lot of methodological details are presented outside of the methods section and with very specific terms which confuse the reader of what is the main point of each section. Each section of a manuscript has a specific role in telling the story and by respecting these roles we ensure that the reader can follow the story and look for the sections he is mostly interested in. I tried to suggest some general rules that I hope can guide the author in future versions of the manuscript. My general suggestion is to simplify the text, reduce the jargon and follow the formal structure suggested in scientific writing to make the readers’ task a bit easier and help them as much as we can in disentangling our objectives and analyses.

6. PLOS authors have the option to publish the peer review history of their article (what does this mean?). If published, this will include your full peer review and any attached files.

Reviewer #1: No

Reviewer #2: No

---

## [Author Response · Author response to Decision Letter 0]

25 Oct 2023

SOME IMPORTANTS NOTES

For this version, you will notice some changes in the numerical outputs. The reason is that I included some observations I had eliminated in my first approach for this problem while working with the entire AVONET dataset. I had cleaned the dataset from missing observations, scanning all variables, but since I am working with a subset of all variables, it is more appropriate to clean using only the variables I am working with. That is why I now have 11,009 species (observations). The results are essentially the same.

REVIEWER #1

- Throughout I think the objective of this research is not emphasized strongly but particularly in the abstract and the opening introduction paragraph.

A/: Thank you so much for the advice. I applied the pertinent changes.

- The abstract states that "This paper aims to evaluate the predictive potential of morphological data to determine the primary lifestyle of birds through a machine learning algorithm." but not why you would want to do that, or how it will help solve various ecological questions. Is the aim just to show that you can? It isn't very surprising that you can predict at some level primary lifestyle from morphology. Or are you trying to show a good method for doing so? What is the measure of being able to determine lifestyle from morphology? With modern machine learning you can build some level of prediction for most things given enough data.

A/: I agree. That problem was addressed according to your observations. The objective clearer that the objective of this work is to assess whether those categories are suitable or not for splitting the avian species according to shared morphological characteristics.

- The abstract goes into too much methodology.

A/: The abstract was changed to offer the minimal amount of methodological information required to understand the overall idea of the paper.

- "The best model generated, for which all 11 morphological traits were used, had an accuracy of 83%.": I don't feel that this is a very good prediction level for something as basic as primary lifestyle given everything has been included. What does a model such as just wing span and beak length/width give you? Of these 11 traits, many are going to be highly correlated, for example body mass has continually been shown to be correlated to wing-span and beak size. Is this model over-fitted? It would be good to see some analysis of number of traits to prediction accuracy - possibly plotted as (x number of traits, y prediction power) for both the test and training set. It would also be good to see some analysis for tuning nrounds/eta and depth (possibly also colsample_bytree,subsample and min_child_weight - from stackoverflow articiles on reducing overfitting).

A/: Regarding this point, I agree with you that 84% appears to be not good at all, but it is important to take into consideration that avian morphospace is extraordinarily variable. Therefore, to achieve 84% accuracy in the classification task should be considered a good prediction. In previous studies, similar but not better levels have been achieved (70-85%, refer to Pigot et al., 2020).

However, to avoid the bias of subjectivity regarding considering or not that accuracy as good or bad, I preferred the use of an objective measure. I used Kappa statistics as one measure of performance, and even in the worst model, it shows a substantial level of agreement between predictions and real categories. Also, I exploit a hypothesis testing procedure in this sense. I used the one-sided test to see if the accuracy of the model is better than the "No Information Rate" (NIR). The No Information Rate is taken to be the largest class percentage in the data. In other words, it is accuracy that could be achieved by always predicting the most frequent class. The P-value here is calculated using a binomial test. It is a one-sided test because I am only interested in whether the accuracy of the model is significantly better than the NIR. The smaller the P-value, the greater the statistical evidence I have to reject the null hypothesis. In this case, for the best model, a P-value of less than 2.2e-16 is extremely small, which provides strong evidence to reject the null hypothesis that the model's accuracy is not better than the NIR. Therefore, I can conclude that my model's accuracy is significantly better than what would be achieved by always predicting the most frequent class.

Regarding feature selection, machine learning algorithms usually take advance of high dimensional data. In one of the models, I even trimmed out three traits, but the performance was inferior to the achieved in the model based in all traits. Also, correlations between variables can be managed by this algorithm, which can undercover and take advantage of complex relations. However, I considered this during the training design. The first model I built was based on Principal components, which are, by definition, non-correlated summary variables. And again, and opposite to what I thought, had the wort performance in this study. I must confess that I expected that one was the best.

Regarding the tunning process, I ran cross validation based in 10-folds for selecting the best combination of hyperparameters, and then I used them for constructing the final model in all cases. The code will be fully available on Github. This is also a measure against overfitting. And regarding overfitting, I also split my dataset in a training and testing datasets for avoiding data leakage (which almost always derives in overfitting problems) and techniques of early stopping during the training process.

- The introduction needs rework to include the motivation behind this work. Also, a better focus on the aim, for prediction what would be an acceptable prediction accuracy (based on what previous work?). What are the tradeoffs for including more traits vs a simpler model?

A/: I worked on the introduction. I tried to explain the elements associated to the model in the materials and methods to ease the lecture. In this kind of model, though, the introduction of more traits is well handled by the algorithm. In fact, each tree constructed does not use all the features. The algorithm chooses combinations of variables that reduces the error. In this kind of approach, I would say that sometimes fewer traits could diminish the performance, because the model has less flexibility to find rules, the selection pool is smaller.

- In the methods it is not clear to me modelling approach has been chosen. What other approaches were considered? Why were they discarded? It is a very step by step procedure of what was done, but I don't really understand why those decisions were made, what trade offs were involved, or the reasoning for the steps. I find it rather technical and difficult to follow, and don't reach the end understanding why choices have been made. Particularly, why only 3 models with these parameters?

A/: It is true that materials and methods are not as clear as I would have desired. I made changes and I hope that it is more appropriate now. Well, from the beginning I knew I should use machine learning. The antecedent of this work showed how the avian morphospace is extraordinarily variable. I selected a tree-based approach because the performance in this kind of study is usually high. I considered random forest too, but the idea of building shallow classification trees in series to iteratively diminish the error seemed to me more suitable. Moreover, extreme gradient boosting is one of the preferred techniques for classification tasks. I included some classical approaches in the paper as contrasts. Regarding the parameters, I used cross-validation techniques for reaching those values. I am sorry if I was not clear about that and if I provided misleading information that makes you think I chose them randomly. For building the three models I made many potential sub models and then by cross-validation I chose the parameters of the best sub model. I included a new figure for depicting visually the pipeline.

- In the results I am not sure that comparing each morphological trait to primary lifestyle is particular interesting or related to the aims of this paper? I think a better stated aim would improve the results. For figure 2 it would be good to include all PCA axis, and possibly also transform the PCA (e.g. log) to make the plot clearer. Table 2 and figure 4 do not add anything. I do not understand the significance of figure 5.

A/: I believe that univariate comparisons, thought as first approach for dissecting each predictor, is something that offers preliminary ideas about how they look for every level of primary lifestyle. I would prefer to keep them on this paper if it were possible. For the PCA figure, I followed your recommendations. Now it is stated as figure 3, and shows the combination between the 5 Principal Components, the density of every component for every lifestyle, and the ellipsoids. Now it is way more informative. Thank you so much for your observation. Regarding the scaling according to a log scale, I believe it is not necessary. All data have been previously scaled and centered. Table 2 offers all the tools for evaluating the performance of models. It has the recall, precision, prevalence, and other useful information to tell where the model performed the best, which could be the flaws, etc. I would prefer to keep it if possible. Figure 4 was eliminated because is true that does not provide anything new. Figure 5 is a nice way of seeing the results of the classification process done for the models. In the row you have the real categories for the birds, and in the columns the predicted ones. So, the diagonal represents correct classification, and the other cells the misclassifications. That way you can analyze where the model went wrong.

- In conclusion it would be good to know how this paper will allow us to better answer ecology questions about birds.

A/: I agree. Your suggestion has been applied to the manuscript.

- "Overall, the results of this research provide valuable insights into the relationship between morphological traits and primary lifestyle in birds and demonstrate the potential for using machine learning models to accurately predict avian lifestyles based on their physical characteristics." - what were the valuable insights?

A/: There were stated the valuable insights. Thank you so much for your observations.

- In general, the paper is very technical with a lot of technical jargin that I found hard to follow and seems to go very deep into how these 3 models were built. But I don't think I really understanding the reasoning for building these three models. I do think there are some interesting questions to answer here and a good paper can be achieved with some better framing, more hand-holding through the methods, and tying the whole paper together with a stronger aim.

A/: I understand your recommendation, and I agree completely. I tried to follow that general guideline in the entire paper. I hope I have succeeded.

REVIEWER #2

MINOR COMMENTS

- I understand your point of view. I eliminated my current degree that appears next to my name. Thank you for the advice.

- In this case, I can only control the font of the manuscript. The pre-print file is automatically generated, and because of that, I change the font in the rest of it.

DETAILED COMMENTS

ABSTRACT

The text was modified according to your recommendations and observations. Thank you so much.

INTRODUCTION

- l 45, what is an angular stone?

A/: Angular stone is a direct translation of Spanish. I understand the confusion and the terms were changed to ‘keystone’, because the previous one is not used in English.

- l 47 increasing interest

A/: The change was implemented.

- l 47-52. Here there is need for a more general introduction, this is too detailed for the broad readership of this journal. A reader that did not study ecology (and even one who did) might find this introduction confusing and too detailed. Start with a few lines of general introduction such as how is a trait defined, difference between trait and functional trait, why are traits important, why are they used in ecological studies, what are the lifestyles that you want to relate the traits to.

A/: I changed the text according to your suggestions. It is true that is unnecessarily complicated for an introduction.

- l 52 I am confused about what is an ecological system (a species, a population, a community). And how do we transition from talking about ecological system in general to birds in specific? (l 53)

A/: I talk about ecological system because you could work at several levels -species, populations, orders, etc. Maybe it is more straightforward to use the term ‘ecological units.’

- l 54 what about reproduction?

A/: Yes, absolutely. Reproduction was included.

- l 59 says “narrow wings perform best for gliding and soaring” I am not sure this is the case. See e.g. work from Pennycuick.

A/ I see the problem here. I should not have generalized that idea. I changed the sentence to “For example, some studies show that for certain birds, narrow wings perform best for gliding and soaring.”

- l 61 I could not follow why a lower metabolic rate limits dispersion ability. There are very large migrating birds who can travel very long distance, and I am not sure why would they have lower dispersion ability. Also dispersion can be of seeds, of microorganisms, of diseases, etc; depending on what is being dispersed, different traits would come into play.

A/ I apologize for that. In general, larger birds tend to have lower metabolic rates than smaller ones. This is because the energy expenditure per unit of body mass decreases with increasing body size. In other words, a larger bird uses more total energy than a smaller bird, but less energy for each gram of its body mass. This lower metabolic rate in larger birds can limit their dispersion ability. Dispersion refers to the distribution of individuals within a population. Birds with lower metabolic rates may not be able to travel as far or as fast as birds with higher metabolic rates. This could limit their ability to disperse and colonize new areas. 

The lower metabolic rate of larger birds can also promote terrestrial foraging strategies. Terrestrial foraging involves searching for food on the ground, as opposed to aerial foraging, which involves searching for food while flying. Terrestrial foraging can be less energy-intensive than aerial foraging, making it a more suitable strategy for birds with lower metabolic rates. An example of this could be the ostrich.

- l 66 to 82 these paragraphs have a lot of jargon (technical language), and by the end of the introduction the reader cannot understand why this study is needed and what the purpose of the study is, which is the main point that the introduction section should fulfill. I would recommend the author to revise the manuscript following general guidelines regarding the structure of a scientific manuscript.

A/ Thank you so much for your recommendation. I modified the Introduction accordingly and I hope the purpose of this work is clear now.

- l 97 a definition of what these lifestyles are is recommended. Ideally you would not want the reader to read another paper before being able to understand yours; on the opposite the author wants to make the reader’s job as easy as possible to make sure they take home the most they can from the paper.

A/ You are absolutely right. A definition of every category was included in the corresponding section of materials and methods.

DICUSSION

- The first paragraph of the discussion should be rewritten, a discussion starts with a summary of what the results of the study are and then slowly puts them in context and finally highlights the relevance of the results for the field, from specific to general, somehow opposite to the introduction.

A/ First paragraph was corrected according to your suggestion. Thank you so much.

---

## [Decision Letter · Decision Letter 1]

17 Nov 2023

Assessing the predictive value of morphological traits on primary lifestyle of birds through the extreme gradient boosting algorithm

PONE-D-23-14687R1

Dear Dr. Madrigal-Roca,

We’re pleased to inform you that your manuscript has been judged scientifically suitable for publication and will be formally accepted for publication once it meets all outstanding technical requirements.

Kind regards,

Vitor Hugo Rodrigues Paiva, Ph.D.

Academic Editor

PLOS ONE

Additional Editor Comments (optional):

Reviewers' comments:

Reviewer's Responses to Questions

**Comments to the Author**

1. If the authors have adequately addressed your comments raised in a previous round of review and you feel that this manuscript is now acceptable for publication, you may indicate that here to bypass the “Comments to the Author” section, enter your conflict of interest statement in the “Confidential to Editor” section, and submit your "Accept" recommendation.

Reviewer #3: All comments have been addressed

Reviewer #4: (No Response)

2. Is the manuscript technically sound, and do the data support the conclusions?

Reviewer #3: Yes

Reviewer #4: Partly

3. Has the statistical analysis been performed appropriately and rigorously? 

Reviewer #3: N/A

Reviewer #4: No

4. Have the authors made all data underlying the findings in their manuscript fully available?

Reviewer #3: Yes

Reviewer #4: Yes

5. Is the manuscript presented in an intelligible fashion and written in standard English?

Reviewer #3: Yes

Reviewer #4: Yes

6. Review Comments to the Author

Reviewer #3: Comments and suggestions were both considered by the authors in detail. I recommend to accept it in the current form.

Reviewer #4: This paper combines bird morphological trait data with a classification of avian lifestyles to ask whether traits can predict lifestyles. It’s a reasonable hypothesis and seems timely in relation to recent interest in trait-based metrics of biodiversity and associated ecological functions. While the result is interesting I think more needs to be done to place it in context and introduce the appropriate caution around the methods.

My first main point is that there should be a clearer statement that previous work (Pigot et al. 2020) used the same dataset and similar machine learning models to test whether the traits predicted diet. The point should be made that machine-learning evidence of trait-based predictions of diet makes the current contribution a relatively minor advance.

Of more concern is that the authors do not seem to consider the limitations of using a machine learning model that could be massively overfitting the data. The way the methods work, the models may be fitting a very convoluted relationship between membership of particular lifestyle categories and a particular trait axis. This risk of over-fitting should be mentioned as a potential risk of machine-learning methods, and ideally the authors should show the relationship between traits and lifestyles described by the models. I suspect that the risk of over-fitting is greater for the lifestyle data than it was for the dietary data (Pigot et al. 2020) because there are even fewer lifestyle categories.

I also don’t think the authors account for the contribution of phylogeny to predictability when they build their training and testing datasets. Previous studies (e.g. Pigot et al. 2020) accounted for this by comparing predictive ability to a null model where traits evolved across the phylogeny. I am not necessarily suggesting that the authors of the current study should take this step, but they should at least carefully acknowledge that their model may overfit the data and overlook the predictive role of phylogeny, pointing to previous methods that have accounted for that criticism and suggesting that further studies are needed to explore these issues.

Minor points:

Why are there two titles? I prefer the shorter second one.

Abstract: “However, there does not exist a previous exploration of the relationship between morphological traits and the primary lifestyle of birds” – change to:

“However, the relationship between morphological traits and the primary lifestyle of birds has not previously been explored”

“avian organisms” sounds like bullshitty jargon. Try just “birds” or “bird species”

“Supplementary dataset 1 of the AVONET project was used” – I would change this to “I asued a global dataset of avian functional traits (AVONET)…”

I would not use terms like “xgboost” in the abstract. The abstract is not the place for off-putting technical jargon. State plainly and explicitly what the models were trying to do, how they differ from other models (and in the methods you can explain that they are called xgboost models).

“for the first time” – don’t use claims of primacy so bluntly in the abstract (or anywhere). Delete this phrase. Let the reader decide.

7. PLOS authors have the option to publish the peer review history of their article (what does this mean?). If published, this will include your full peer review and any attached files.

Reviewer #3: No

Reviewer #4: No

---

## [Editor Report · Acceptance letter]

21 Dec 2023

PONE-D-23-14687R1 

PLOS ONE

Dear Dr. Madrigal-Roca, 

I'm pleased to inform you that your manuscript has been deemed suitable for publication in PLOS ONE. Congratulations! Your manuscript is now being handed over to our production team.

Kind regards, 

on behalf of

Dr. Vitor Hugo Rodrigues Paiva 

Academic Editor

PLOS ONE